# Multidimensional recurrence quantification analysis of human-metronome phasing

**Caitrín Hall**[ID]*, **Ji Chul Kim**[ID], **Alexandra Paxton**[ID]

Department of Psychological Sciences, University of Connecticut, Storrs, CT, United States of America

* caitrin.hall@uconn.edu

## Abstract

Perception-action coordination (also known as sensorimotor synchronization, SMS) is often studied by analyzing motor coordination with auditory rhythms. The current study assesses phasing—a compositional technique in which two people tap the same rhythm at varying phases by adjusting tempi—to explore how SMS is impacted by individual and situational factors. After practice trials, participants engaged in the experimental phasing task with a metronome at tempi ranging from 80–140 beats per minute (bpm). Multidimensional recurrence quantification analysis (MdRQA) was used to compare nonlinear dynamics of phasing performance. Varying coupling patterns emerged and were significantly predicted by tempo and linguistic experience. Participants who successfully phased replicated findings from an original case study, demonstrating stable tapping patterns near in-phase and antiphase, while those unsuccessful at phasing showed weaker attraction to in-phase and antiphase.

**Data Availability Statement:** All raw data, code, protocols, and other methods materials are available from the Open Science Framework

## Introduction

Perception-action coordination (also called sensorimotor synchronization, SMS) occurs as people coordinate overt movements with a rhythmic stimulus [1–3]. This is typically studied using tasks in which participants are instructed to coordinate with auditory rhythms, which requires explicit intention to synchronize. We aim to explore coordination when participants are instructed to execute controlled *desynchronization* with a rhythmic stimulus. This task, called phasing, is inspired by a musical technique investigated in a case study of professional percussionists [4]. Our controlled examination of phasing provides a conceptual replication and extension of the case study.

The simplest synchronization task is coordinating rhythmic finger movements with a metronome. Within such contexts, phase dynamics are largely involuntary, while tempo change is under voluntary control [5–7]. People display two stable phase relationships: in-phase and anti-phase [5,8]. Individuals also display substantial tempo flexibility, coordinating from 17 bpm to 150–170 bpm for anti-phase tapping [8–10] and 300–400 bpm for in-phase tapping [10–13]. Prior research suggests it becomes difficult to coordinate with a metronome beyond these rate limits.

Rhythmic coordination is also impacted by individual factors, such as musical and linguistic experience. Previous findings suggest musicians are better able to synchronize with auditory

database (https://osf.io/4uhtb/?view_only=
f8b5a974b1b54bff95abbc41a8342f5e).

**Funding:** This work was financially supported by
the Peter and Carmen Lucia Buck (PCLB)
Foundation Undergraduate Research Grant from
the Connecticut Institute for Cognitive and
Behavioral Sciences, which was awarded to C. Hall.
The funders had no role in the study design, data
collection and analysis, decision to publish, or
preparation of the manuscript.

**Competing interests:** The authors have declared
that no competing interests exist.

stimuli than nonmusicians [14,15]. Additionally, an association between increased inhibitory control and SMS abilities in bilinguals [16–18] indicates multilingualism may improve rhythmic coordination. Taken together, perception-action coupling is shaped by individual differences and task demands.

## Phasing

Phasing involves musicians shifting in and out of synchrony with one another while performing the same rhythm at different tempi. For example, in the musical composition "Drumming" [19], two percussionists begin by drumming synchronously. One drummer then attempts to maintain the original tempo while the other accelerates, gradually desynchronizing. Eventually, after tapping the same rhythm at different tempi, the partners approach synchrony again and return to unison by tapping at identical tempi one cycle apart.

During "Drumming," the professional percussionists reported feeling pulled toward synchronization despite intentions to play at independent tempi [20]. To understand this, Schutz [4] conducted a case study of the percussionists performing "Drumming" (Fig 1). In theory, the accelerating and steady drummers should perform independently. In reality, however, they exhibit a push-and-pull pattern, unable to overcome the tendency toward interpersonal coupling. Inspired by Schutz's case study, the current work investigates the evolution of coordinative behaviors during phasing with a metronome among non-expert populations using *recurrence quantification analysis* (or *auto-recurrence quantification analysis*; RQA).

## Quantifying phasing with a novel application of RQA

RQA is a nonlinear time series analysis [21] that has been applied to coordination research [22,23] to quantify a single dynamical system over time [24]. Conceptually, RQA uncovers the structure of a system that changes over time through examining the patterns of repetition of a single variable of interest over time. RQA uses only one observable to describe the dynamics of multiple interdependent variables of a system, as the interacting components of a dynamical system can be uncovered from a single variable [24,25]. The idea is to reconstruct the phase space of the dynamical system, recover the trajectory of the system through that phase space, and then plot the system's trajectory against itself to identify patterns of recurrent (or repeating) discrete states or regions of the state space. This creates a recurrence plot (RP), from which various metrics may be derived to quantify the dynamics of the system as a whole [25].

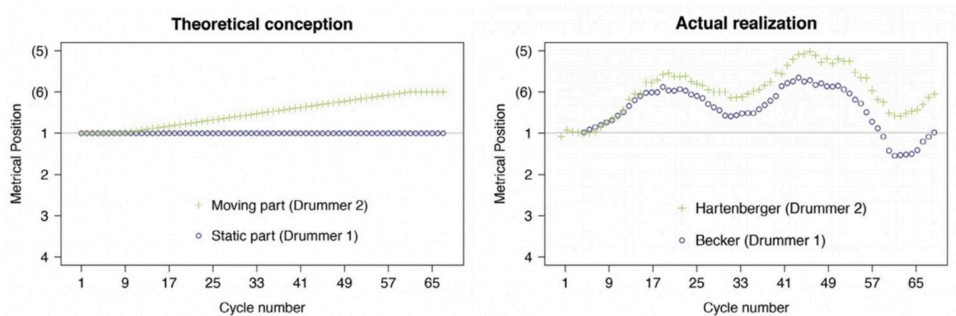

**Fig 1. Theoretical versus actual phasing performance.** *Left*: How phasing would occur, in theory, if partners were uninfluenced by each other's performance. The static part (blue) remains constant, while the moving part (green) steadily shifts its metrical position. *Right*: Actual performance of "Drumming" by Becker and Hartenberger. The "static" part actually varies along with the moving part during phasing. In other words, the drummer who intended to maintain the original tempo was unable to do so; instead, this drummer unintentionally *increased* and *decreased* their tempo along with the phasing drummer. (Figure reproduced with permission from Schutz [4]).

RQA has also been adapted to analyze systems with more than one measured variable and to examine the dynamics of coupled systems, including *cross-recurrence quantification analysis* (CRQA) and *multidimensional recurrence quantification analysis* (MdRQA). CRQA quantifies the coevolution of two distinct but interacting systems; in other words, CRQA captures the shared trajectories of two separate univariate systems. On the other hand, MdRQA analyzes a single system captured by two or more measured variables; that is, instead of the variables belonging to distinct time series, the variables are different dimensions of the same time series. Thus, MdRQA quantifies the auto-recurrence properties of a single multidimensional or multivariate system [25]. Readers interested in more detailed conceptual and mathematical explanations of RQA and its extensions are encouraged to consult Carello and Moreno [24] and Wallot et al. [25].

We quantify phasing by analyzing *relative phase* ($\psi$), which measures the angle between the two phasing signals in degrees. However, to date, the extensions of RQA cannot adequately handle such circular data. Because RQA-based methods operationalize similarity through the revisiting of a similar state within a given radius, RQA detects an apparent discontinuity between 359˚ and 0˚; it calculates an angular difference of 359˚ even though going from 359˚ to 0˚ is a difference of only 1˚ in terms of $\psi$. Our conceptual framework necessitates that the shift from 359˚ to 0˚ be interpreted as the same $\psi$ change as that when moving from 0˚ to 1˚.

For the current work, we therefore created a novel circular extension of MdRQA by decomposing the relative phase signal into its *x*- and *y*-coordinates and using MdRQA to analyze them together as a multidimensional signal of the same system. Because relative phase is inherently a relational measure—in our case, a measure that necessarily accounts for the positions and relations of two phasing signals—this means that the *x*-- and *y*-coordinate values must be considered coequal and non-separable parts of a single multidimensional system. As a result, CRQA would be unsuitable for the current study, as it would treat the *x*- and *y*-coordinates as two separate but interacting systems. MdRQA, on the other hand, treats both coordinates as part of the same system, allowing us to accurately capture the system revisitations (or approximate revisitations) through specific *x,y* regions of the relative phase. Importantly, the choice to create and implement this novel extension of MdRQA to analyze these data came after the data were collected, as it provided for a more complete accounting of the nonlinear dynamics of the system. More information on our motives and implementation are available in the "Data Analysis" section.

In the present study, we use MdRQA to characterize multiple dimensions of a singular relative phase variable. We also use region-based MdRQA, which is the same as general MdRQA mathematically and methodologically except that it parses the RP into subsections and analyzes each subsection independently. Thus, region-based MdRQA allows us to quantify different relative phase relations separately. Like RQA, both general and region-based MdRQA yield metrics that quantify the structure, patterns, and stability of a system's evolution. Here, we target percent recurrence (%REC), percent determinism (%DET), and maxline (MAXL). %REC is the density of recurrent states of the system across time and is proportional to the inverse of the system's noise. %DET captures recurrent patterns of states across time, quantifying the system's predictability. MAXL is the length of the longest run of consecutive recurrent states, corresponding to the system's attractor strength [25–27].

## Present study

Inspired by Schutz [4], the present study evaluates phasing among non-expert participants with more varied musical and linguistic backgrounds. To maximize feasibility, we simplify Schutz's task in two ways: Participants phase with a metronome instead of a partner, and the

metronome produces an isochronous rhythm. Furthermore, we narrow the tempo range known to permit successful antiphase tapping to 80–140 bpm. We measure phasing by calculating $\psi$ between participants' taps and metronome ticks [28–29]. Using MdRQA to analyze $\psi$, we quantify the human-metronome system's repetitiveness (%REC), predictability (%DET), and attractor strength (MAXL). Consistent with Schutz [4] and coordination dynamics literature [5,8], we hypothesize that participants will demonstrate stable tapping near in-phase and antiphase, indicated by higher metrics during those periods.

Successfully phasing requires attending to the metronome while voluntarily adopting a different tempo than the metronome [7]. Ignoring the metronome would result in not knowing when to stop phasing, while an inability to voluntarily adopt a different tempo would cause failure to escape the in-phase attractor—a $\psi$ near 0 throughout the task. Based on evidence of bilingual speakers' increased selective attention [30–32], we hypothesize that multilingual participants may be able to simultaneously attend to the metronome while adopting a different tempo. Monolingual participants may experience stronger coupling with the metronome. This would result in greater metrics for monolingual speakers. Furthermore, considering tapping rate limits, we expect the middle range of our selected tempi (100–120 bpm) to yield the most structured phasing performance for all participants, resulting in higher metrics. This prediction aligns with the preferred tempo range [33,34], which is the zone at which tempo perception is optimal—not so fast that individual pulses appear to fuse together and not too slow that individual pulses sound isolated.

## Hypotheses

H1. Middle range tempi (100–120 bpm) will yield the most structured phasing for all participants. *Operationalization*: This will lead to higher %REC, %DET, and MAXL for trials between 100–120 bpm than trials above or below that range.

H2. Participants will demonstrate more stable tapping near antiphase when compared to other nonsynchronous relative phases, due to the increased general stability of antiphase relations when tapping. *Operationalization*: This will be reflected by higher %REC, %DET, and MAXL during antiphase.

H3. Monolingual participants will demonstrate stronger coupling with the metronome than multilingual speakers, due to multilinguals' improved selective attention abilities compared with monolinguals. *Operationalization*: This would result in higher %REC, %DET, and MAXL for monolingual speakers overall.

As noted above, we chose to use MdRQA for the analysis after designing the study and collecting the data. This necessarily means that the operationalization of each hypothesis in terms of MdRQA metrics came later, but these operationalizations nevertheless reflected the *a priori* hypotheses that the study was designed to test.

## Materials and methods

### Participants

Twenty-five undergraduates (17 females, 8 males; 18–21 years, *M* = 18.7 years) at the University of Connecticut were recruited via the Psychological Sciences Participant Pool and received course credit for completing the experiment. All subjects reported normal hearing abilities and no neurological health complications. Thirteen (9 females, 4 males) reported being monolingual, and 12 (8 females, 4 males) reported being multilingual. Six (5 females, 1 male)

reported experience playing one or more instruments for at least one year, while 19 (12 females, 7 males) reported no experience playing an instrument.

**Sample size.** As is customary in the field, results with $p < 0.05$ are considered significant. Experimental studies typically require 15–30 participants for adequate statistical power, which led us to recruit 25 non-expert participants for an exploratory study. We did not recruit based on a planned group comparison.

**Ethics statement.** The University of Connecticut Institutional Review Board approved of this study. The IRB-approved study protocol title is "Cognitive/Behavioral Investigation of Music Performance." Participants provided written consent to participate in the study.

## Procedure

The procedure lasted approximately one hour. First, participants completed a demographics survey. This information was collected for IRB purposes. We did not have *a priori* plans to analyze anything from the survey except for musical and linguistic experience. Next, participants were introduced to the task with audio and video demonstrations (created using custom MATLAB code [35]). Tapping data were collected with a Roland HandSonic HPD-20 Digital Hand Percussion Controller [36]. After two practice sessions (see S1 Appendix for description), participants advanced to the experiment. One experimental trial consisted of the following human-metronome phasing method: The metronome's tempo was programmed to tick at one of the seven tempi ranging from 80–140 bpm in 10-bpm increments. The metronome maintained its original tempo throughout the duration of each trial, which lasted a maximum of two minutes.

Participants were instructed to begin by tapping in synchrony with the metronome for several beats. This allowed participants to adjust to the tempo of the current trial. At the sound of a warning signal (i.e., a short, high-pitched beep), participants began phasing (see section "Phasing") with the metronome. Here, the phasing process entailed participants increasing their tapping rate slightly while the metronome continued ticking at its original tempo. Participants were instructed to maintain their new tempo until they resynchronized with the metronome. Because the participant and metronome played at different (ideally, constant) tempi, resynchronization would happen naturally as the participant "lapped" the metronome. This can be thought of as two people running around a track at different speeds: The faster runner will gradually shift farther ahead of the slower runner until, eventually, the faster runner is one whole lap ahead of the slower runner. At that moment, both runners would be instantaneously resynchronized.

When participants resynchronized with the metronome, they were instructed to revert to the original tempo of the metronome and tap in synchrony for several beats before stopping. The period between participants' initial desynchronization with the metronome and their resynchronization after increasing their tempo was considered one round of phasing (also referred to as one phasing lap). Participants were informed that the goal was to complete exactly one round of phasing per trial. After the drum pad detected several seconds of no tapping input, the experiment automatically continued to the next trial. Each of the seven tempi was presented three times per participant in randomized order, totaling 21 trials per participant; this allowed for a more robust sampling of the per-tempi variability than presenting it only once. The monitor displayed the total number of trials remaining.

## Data inclusion criteria

Overall, our dataset included a total of 524 trials: 25 participants each completed 21 trials, three at each of the seven tempi, minus one trial that terminated early because the participant did not tap a single time. This trial was removed from our analyses.

Due to the challenging nature of the task, only 38% of trials successfully completed one round of phasing ("Successful Trials"). A further 41% completed more than one round, lapping the metronome multiple times ("Unsuccessful Trials"). The remaining 21% did not complete any rounds, resulting in zero laps around the metronome ("Incomplete Trials"). Incomplete Trials still contained tapping data and thus were able to be analyzed using MdRQA; participants simply never resynchronized with the metronome during Incomplete Trials and thus failed to execute the phasing task as instructed. Outliers were included in analyses.

All trial types were analyzed with general MdRQA, and only Successful and Unsuccessful Trials were additionally analyzed with region-based MdRQA. Analyses of Unsuccessful and Incomplete Trials are exploratory because—given the simplified task and earlier piloting—we did not anticipate such a high level of noncompliance. Music experience was excluded as a predictor because we did not achieve sufficient variability in our sample. However, music experience data is provided in participants' data files and in S2 Appendix.

## Data analysis

**Open materials, data, and code.** In keeping with the principles of open science, we have made our materials, data, and code for the current project fully available. Study protocols, raw tapping data (stored as MATLAB files), MdRQA analysis code (in MATLAB) [25], and descriptive and inferential statistics (in R) are available at our OSF project page: https://osf.io/4uhtb/?view_only=f8b5a974b1b54bff95abbc41a8342f5e. We used a variety of R libraries for data manipulation, visualization, and analysis, with specific implementation of each included in the open project code: ggplot2, viridis, emmeans, dplyr, tidyverse, rstatix, qwraps2, lme4, GGally, lmerTest, pander, sjPlot, and ggpubr [37–50].

**Relative phase ($\psi$) calculation.** Relative phase ($\psi$) measures the difference between the time at which a participant taps and the nearest metronome tick, yielding the phasing relationship between participant and metronome. This was calculated as follows:

$$\psi = -\frac{2\pi dt}{T}$$

$dt$ = participant's tap time–nearest metronome tick time
$T$ = metronome's interbeat intervalwhere $dt$ is the difference between the participant's tap time and the nearest metronome tick time and $T$ is the metronome's interbeat interval. The negative sign yields positive $\psi$ when the participant taps ahead of the metronome and negative values when the participant lags behind. Thus, $\psi$ increases when phasing is performed by tapping ahead of metronome ticks.

**General MdRQA.** While $\psi$ was the only independent variable measured in the human-metronome system, these circular data were not well suited for RQA because of the apparent discontinuity in $\psi$ when shifting from 359˚ to 0˚; while $\psi$ changes by only 1° during this transition, RQA calculates an angular difference of 359°, which is conceptually incorrect for our purposes. We resolved this by creating a novel circular application of RQA thanks to MdRQA.

For the circular application of MdRQA, we convert $\psi$ into a 2-dimensional variable by wrapping $\psi$ values to range from zero to $2\Psi$; we then decompose each tap into its $x$- and $y$-coordinates by taking the cosine and sine, respectively. This converts the one-dimensional circular operationalization of relative phase into a two-dimensional representation of the same data—the $x$- and $y$-coordinates of circular relative phase as multivariate measurements of a single human-metronome system. These data are therefore more consistent with the principles of MdRQA (which captures the dynamics of a single multidimensional system) rather than those of CRQA (which captures the shared trajectories of two univariate systems).

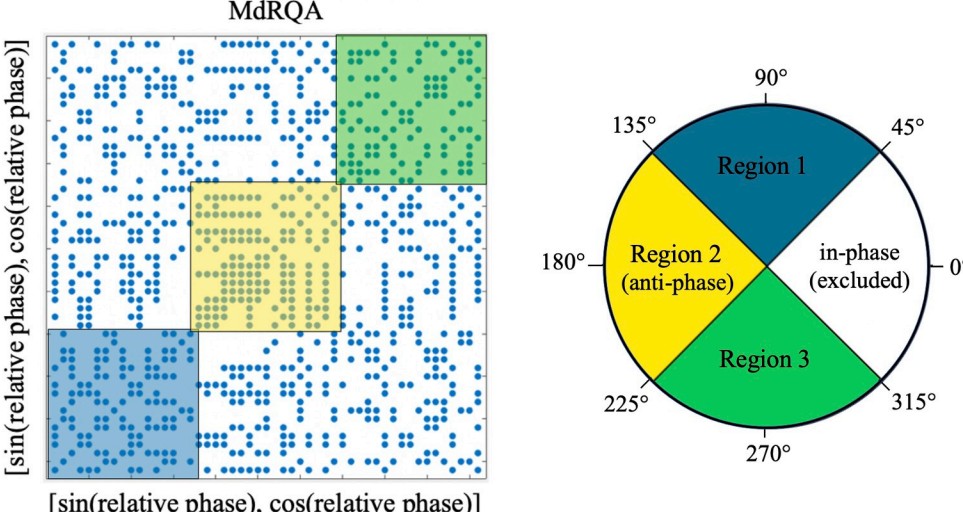

**Fig 2. Sample multidimensional recurrence plot (MdRP) produced for one phasing trial.** (A) How subsections were created from the MdRP to conduct region-based analysis. Each sample in the time series contains a vector of *x*- and *y*-coordinates of ψ. Recurrences or revisitations of the same multidimensional space (within a given tolerance) between the two time series (i.e., the same time series represented on the *x*-axis and *y*-axis) are plotted as points on the MdRP. (B) How participants' taps were assigned to regions 1, 2, and 3 based on circular ψ values. As described in "Region-Based MdRQA," synchronous taps were excluded from region-based MdRQA to effectively compare Successful and Unsuccessful Trials".

While categorical CRQA would have been appropriate for comparing ticks of the metronome with taps of the participant (i.e., two univariate discrete time series), the decomposition of relative phase into two component parts creates two signals from the same system. However, simply conducting categorical CRQA on the metronome ticks and participant taps would not uncover the richness of the relative phase data because of the rigidity of categorical RQA techniques and the relatively short time series.

To capture the dynamics of the continuous relative phase variable, continuous MdRQA was conducted on the two-dimensional vector comprising the *x*- and *y*-coordinates of each circular relative phase value. From there, conducting MdRQA generates a recurrence plot (RP) that describes repetitions of the system's values within a given tolerance across the phase space (Fig 2A). Here, a point on the recurrence plot indicates that a given ψ is repeated within a given radius in the time series. Recurrence metrics are then calculated from the RP. For phasing, higher %REC indicates repetition of the same ψ throughout the trial. Higher %DET means the ψ trajectory is more predictable, and higher MAXL corresponds to stronger attraction to a particular ψ.

MdRQA was performed in MATLAB [25,35]. As specified in the MATLAB files within our linked OSF page, there are five parameters that must be specified when running MdRQA: number of embedding dimensions (EMB), delay (DEL), type of norm by which the phase space is normalized (NORM), radius size (RAD), and whether the data should be z-scored (ZSCORE). We used only a single embedding dimension (EMB = 1) because we did not need to reconstruct the phase space for the current data; both dimensions of the relative phase data (i.e., the x- and y-coordinates) are represented within the dataset. As a result, we used the default delay value for unembedded data (DEL = 1). We did not normalize the phase space (NORM = 'non') or the data (ZSCORE = 0) because both time series exist naturally within the same scale. A small radius (RAD = 1) was chosen because the data is not highly deterministic and thus requires a radius not too close to zero to capture recurrence.

**Region-based MdRQA.**   Using region-based MdRQA allowed us to compare %REC, %DET, and MAXL for the human-metronome system at different relative phase regions. Specifically, we were able to assess differences in recurrence, predictability, and attractor strength during nonsynchronous regions between the initial desynchronization and final resynchronization with the metronome. In other words, we focused on the system's trajectory after it moved from synchrony, passed through antiphase, and approached synchrony again, rather than focusing on in-phase dynamics. Region-based MdRQA was conducted on both Successful and Unsuccessful Trials. We parsed taps into three non-overlapping regions based on circular $\psi$ values: $45° \leq \psi < 135°$ were region 1, $135° \leq \psi < 225°$ were region 2, and $225° \leq \psi < 315°$ were region 3 (Fig 2B). MdRQA metrics were calculated for each region separately using the same parameter values that were used for general MdRQA. Synchronous taps were excluded from region-based MdRQA to effectively compare Successful and Unsuccessful Trials: Although Unsuccessful Trials passed through multiple synchronous points while phasing, there were insufficient points within the synchronous region to calculate metrics.

**Tempo.**   Because our hypotheses included predicted nonlinearities in performance across tempi, we binned tempo to create a categorical variable with three levels: *lower-* (80 and 90 bpm), *middle-* (100, 110, and 120 bpm), and *upper-* (130 and 140 bpm) *range* tempi.

**Model specifications.**   Using the lme4 library in R [37], we created two classes of linear mixed-effects models: one to assess general MdRQA and another for region-based MdRQA. We created a separate equation for each of the three dependent variables (i.e., %REC, %DET, and MAXL) for general and region-based analyses, totaling six models. For general MdRQA, we used Incomplete Trials, monolingual language experience, and middle-range tempi as reference categories; for region-based MdRQA analyses, we used Successful Trials, monolingual language experience, middle-range tempi, and region 1 as the reference categories. We used deviation coding for all categorical variables (see S3 Appendix) [51]. For all models, participant identity was included as a random effect to control for multiple trials per participant.

## Results and discussion

As described in the "Model Specifications" section (above), we analyzed data with two classes of linear mixed effects models for each MdRQA metric. Statistically significant and nonsignificant results are presented in tables; only statistically significant results ($p < 0.05$) are discussed in the text. For readability and clarity within the text, all statistics—including effect sizes—are presented only in the tables. We present results of our *a priori* hypotheses before turning to our exploratory analyses and considering future directions.

### Hypothesis 1

In H1, we predicted that tempi from 100 to 120 bpm would yield the most structured tapping data. Because H1 does not consider phasing regions (e.g., synchrony, antiphase), we use general MdRQA results to assess H1. Summary statistics for general MdRQA are presented in Table 1. Results of the statistical analyses for H1 are available in Table 2.

**Table 1. Descriptive statistics for general MdRQA for successful, unsuccessful, and incomplete trials.**

| Metric | Successful (N = 200) | Unsuccessful (N = 213) | Incomplete (N = 111) |
|---|---|---|---|
| %REC | M = 42.96, SD = 1.01 | M = 42.94, SD = 0.85 | M = 43.22, SD = 0.74 |
| %DET | M = 66.48, SD = 2.42 | M = 66.59, SD = 2.42 | M = 66.93, SD = 2.36 |
| MAXL | M = 7.64, SD = 1.81 | M = 7.95, SD = 1.80 | M = 8.06, SD = 1.51 |

**Table 2. Linear mixed effect model results for general MdRQA with trial type, language experience, and tempo range as predictors of %REC, %DET, and MAXL.**
Each MdRQA metric was predicted using a separate model. Marginal and conditional $R^2$ for each model included below each model label. Effect sizes provided as standardized estimates (ß) for statistically significant predictors. As noted in the Model Specifications section, the use of deviation coding yields $k$-1 variables that test the difference between a given level of the categorical variable and the reference level. General MdRQA models used the incomplete trials, monolingual language experience, and middle tempi as reference categories; for more on mathematical interpretation of deviation-coded interaction terms, see Barr et al. (2013) [51].

| **%REC** Marginal R2 = 0.051, Conditional R2 = 0.112 | | | | | | |
|---|---|---|---|---|---|---|
| Predictors | Estimate | SE | df | t-value | p-value | ß |
| (Intercept) | 43.044 | 0.070 | 42.98 | 610.665 | < 0.0001 | – |
| Successful trials (v. Incomplete) | -0.265 | 0.152 | 499.96 | -1.75 | 0.081 | – |
| Unsuccessful trials (v. Incomplete) | -0.178 | 0.154 | 467.95 | -1.157 | 0.248 | – |
| Multilingual language exp. (v. Monolingual) | 0.128 | 0.130 | 31.63 | 0.987 | 0.331 | – |
| Upper tempo range (v. Middle tempo range) | 0.044 | 0.096 | 488.68 | 0.461 | 0.645 | – |
| Lower tempo range (v. Middle tempo range) | -0.129 | 0.136 | 499.71 | -0.944 | 0.346 | – |
| Successful trials x Language exp. | 0.003 | 0.257 | 494.26 | 0.013 | 0.990 | – |
| Unsuccessful trials x Language exp. | -0.137 | 0.265 | 449.04 | -0.518 | 0.605 | – |
| Successful trials x Upper tempo | 0.040 | 0.232 | 494.29 | 0.174 | 0.862 | – |
| Unsuccessful trials x Upper tempo | 0.218 | 0.247 | 499.25 | 0.88 | 0.380 | – |
| Successful trials x Lower tempo | -0.365 | 0.392 | 501.10 | -0.93 | 0.353 | – |
| Unsuccessful trials x Lower tempo | -0.409 | 0.392 | 502.76 | -1.044 | 0.297 | – |
| Language exp. x Upper tempo | -0.075 | 0.192 | 488.68 | -0.392 | 0.695 | – |
| Language exp. x Lower tempo | -0.428 | 0.204 | 491.07 | -2.097 | 0.037 | -0.109 * |
| Successful tr. x Lang. exp. x Upper tempo | -0.069 | 0.463 | 494.29 | -0.149 | 0.882 | – |
| Successful tr. x Lang. exp. x Lower tempo | -0.342 | 0.404 | 498.55 | -0.848 | 0.397 | – |
| Unsuccessful tr. x Lang. exp. x Upper tempo | -0.235 | 0.495 | 499.25 | -0.474 | 0.636 | – |

| **%DET** Marginal R2 = 0.033, Conditional R2 = 0.049 | | | | | | |
|---|---|---|---|---|---|---|
| Predictors | Estimate | SE | df | t-value | p-value | ß |
| (Intercept) | 66.546 | 0.161 | 58.88 | 412.154 | < 0.0001 | – |
| Successful trials (v. Incomplete) | -0.010 | 0.410 | 457.77 | -0.024 | 0.981 | – |
| Unsuccessful trials (v. Incomplete) | 0.081 | 0.412 | 389.10 | 0.198 | 0.843 | – |
| Multilingual language exp. (v. Monolingual) | 0.404 | 0.286 | 38.77 | 1.414 | 0.165 | – |
| Upper tempo range (v. Middle tempo range) | 0.301 | 0.266 | 492.01 | 1.132 | 0.258 | – |
| Lower tempo range (v. Middle tempo range) | -0.419 | 0.376 | 505.54 | -1.115 | 0.266 | – |
| Successful trials x Language exp. | -0.043 | 0.694 | 438.87 | -0.062 | 0.950 | – |
| Unsuccessful trials x Language exp. | -0.689 | 0.705 | 358.87 | -0.977 | 0.329 | – |
| Successful trials x Upper tempo | -0.267 | 0.641 | 500.43 | -0.416 | 0.677 | – |
| Unsuccessful trials x Upper tempo | -0.694 | 0.682 | 505.53 | -1.018 | 0.309 | – |
| Successful trials x Lower tempo | 0.580 | 1.080 | 506.71 | 0.537 | 0.591 | – |
| Unsuccessful trials x Lower tempo | 0.807 | 1.078 | 507.00 | 0.748 | 0.455 | – |
| Language exp. x Upper tempo | -0.395 | 0.531 | 492.01 | -0.744 | 0.457 | – |
| Language exp. x Lower tempo | -0.526 | 0.566 | 494.83 | -0.93 | 0.353 | – |
| Successful tr. x Lang. exp. x Upper tempo | 1.549 | 1.281 | 500.43 | 1.209 | 0.227 | – |
| Successful tr. x Lang. exp. x Lower tempo | 1.818 | 1.114 | 504.98 | 1.632 | 0.103 | – |
| Unsuccessful tr. x Lang. exp. x Upper tempo | 0.609 | 1.364 | 505.53 | 0.446 | 0.655 | – |

| **MAXL** Marginal R2 = 0.073, Conditional R2 = 0.215 | | | | | | |
|---|---|---|---|---|---|---|
| Predictors | Estimate | SE | df | t-value | p-value | ß |
| (Intercept) | 7.942 | 0.168 | 33.58 | 47.34 | < 0.0001 | – |
| Successful trials (v. Incomplete) | -0.196 | 0.283 | 506.36 | -0.691 | 0.490 | – |
| Unsuccessful trials (v. Incomplete) | 0.247 | 0.291 | 504.52 | 0.851 | 0.395 | – |

*(Continued)*

**Table 2.** (Continued)

| | | | | | | |
|---|---|---|---|---|---|---|
| Multilingual language exp. (v. Monolingual) | 0.256 | 0.320 | 27.82 | 0.8 | 0.430 | – |
| Upper tempo range (v. Middle tempo range) | 0.612 | 0.176 | 486.74 | 3.477 | 0.001 | 0.16 *** |
| Lower tempo range (v. Middle tempo range) | -0.221 | 0.252 | 493.47 | -0.877 | 0.381 | – |
| Successful trials x Language exp. | -0.371 | 0.481 | 506.92 | -0.771 | 0.441 | – |
| Unsuccessful trials x Language exp. | 0.306 | 0.501 | 500.55 | 0.61 | 0.542 | – |
| Successful trials x Upper tempo | 0.645 | 0.426 | 489.77 | 1.511 | 0.131 | – |
| Unsuccessful trials x Upper tempo | 0.333 | 0.456 | 492.91 | 0.731 | 0.465 | – |
| Successful trials x Lower tempo | -0.732 | 0.724 | 494.13 | -1.012 | 0.312 | – |
| Unsuccessful trials x Lower tempo | -0.898 | 0.724 | 495.49 | -1.239 | 0.216 | – |
| Language exp. x Upper tempo | -0.341 | 0.352 | 486.74 | -0.97 | 0.333 | – |
| Language exp. x Lower tempo | -0.696 | 0.376 | 488.35 | -1.854 | 0.064 | -.09 . |
| Successful tr. x Lang. exp. x Upper tempo | 0.516 | 0.853 | 489.77 | 0.605 | 0.545 | – |
| Successful tr. x Lang. exp. x Lower tempo | 0.165 | 0.745 | 492.43 | 0.221 | 0.825 | – |

Overall, our general MdRQA findings failed to support H1. Neither %REC (Fig 3A) nor % DET (Fig 3B) were significantly greater for middle tempi than for lower and upper tempi. MAXL (Fig 3C) did not significantly differ for middle versus lower tempi, but MAXL was significantly greater for upper tempi than for middle tempi. The attractor strength of the human-metronome system grew as tempo increased from 100–140 bpm, making it more difficult for participants to decouple from the metronome and successfully achieve phasing.

Our findings might imply that our tempo selection was too narrow. Based on previous research, participants tapping at a comfortable tempo should experience a stronger pull toward attractors, resulting in more structured, repetitive tapping as reflected in greater MdRQA metrics. Our finding that participants produced the most structured data (i.e., MAXL) at upper range tempi suggested that this may have been a more comfortable tempo for phasing than lower and middle range tempi. According to previous literature, the average natural preferred tempo for people falls around 120 bpm [33,34]. To account for the difficulties of phasing (versus tapping in-phase with a metronome), we chose to center our tempo range at 110 bpm to make the task more manageable for participants. Our results, however, suggest that future phasing studies should test faster tempi.

## Hypothesis 2

H2 predicted that, overall, participants would demonstrate more stable tapping during antiphase (i.e., region 2) than during other relative phases (i.e., regions 1 and 3). This would have been supported by higher %REC, %DET, and MAXL during region 2. Because H2 pertains to phasing regions, we use region-based MdRQA results to assess our prediction. Summary statistics for region-based MdRQA are presented in Table 3. Results of the statistical analyses for H2 are available in Table 4. The significant findings related to H2 include the main effect of region on %REC, %DET, and MAXL (Fig 4); interactions between trial type and region on both %DET (Fig 4B) and MAXL (Fig 4C); and interactions among language experience, tempo, and region on both %DET (Fig 5B) and MAXL (Fig 5C).

The main effect of region on %REC revealed that region 2 had significantly lower %REC than region 1, suggesting that antiphase was noisier than desynchronization and failing to support H2. Instead, region 1 had significantly greater %DET and MAXL values than region 3. In other words, the relative phase region just before returning to synchrony was the least predictable and exhibited the weakest coupling between human and metronome.

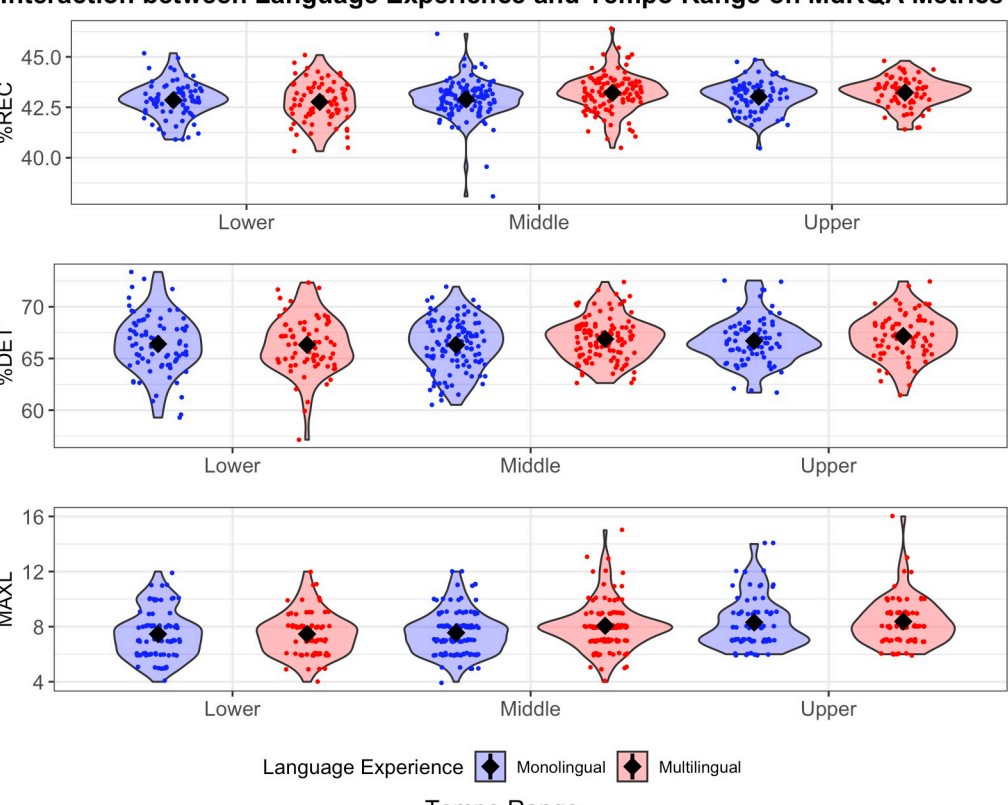

**Fig 3. Effects of language experience (red: Multilingual; blue, monolingual) and tempo range on MdRQA metrics: %REC (Panel A, top), %DET (Panel B, middle), and MAXL (Panel C, bottom).** The metronome ranged from 80–140 bpm. In our linear mixed effects model, we binned tempo into the following categories: lower (80–90 bpm), middle (100–120 bpm), and upper (130–140 bpm). The violin plots depict the probability density of the binned tempo data at different %REC, %DET, and MAXL values. The mean value is indicated by a diamond near the center of each violin. Results of statistical analyses are located in Table 2.

The interaction between trial type and region on %DET supported H2: Both Successful and Unsuccessful Trials yielded the highest %DET during region 2 and smallest during region 3. This meant that the human-metronome system exhibited the most predictable tapping pattern during region 2, as anticipated. Successful Trials were generally more predictable than Unsuccessful Trials.

**Table 3. Descriptive statistics for region-based MdRQA for successful and unsuccessful trials.**

| Region | Metrics | Successful (N = 200) | Unsuccessful (N = 213) |
|---|---|---|---|
| 1 | %REC | M = 24.84, SD = 16.20 | M = 17.00, SD = 19.81 |
| | %DET | M = 23.86, SD = 31.00 | M = 6.63, SD = 20.17 |
| | MAXL | M = 1.58, SD = 1.35 | M = 0.65, SD = 0.79 |
| 2 | %REC | M = 25.78, SD = 14.17 | M = 14.10, SD = 21.20 |
| | %DET | M = 26.29, SD = 29.98 | M = 10.26, SD = 26.06 |
| | MAXL | M = 1.76, SD = 1.26 | M = 0.54, SD = 0.82 |
| 3 | %REC | M = 22.06, SD = 18.45 | M = 16.70, SD = 21.29 |
| | %DET | M = 14.35, SD = 26.67 | M = 3.92, SD = 16.10 |
| | MAXL | M = 1.04, SD = 0.91 | M = 0.50, SD = 0.63 |

**Table 4. Linear mixed effect model results for region-based MdRQA with trial type, language experience, tempo range, and region as predictors of %REC, %DET, and MAXL.** Each MdRQA metric was predicted using a separate model. Marginal and conditional $R^2$ for each model included below each model label. Effect sizes provided as standardized estimates (ß) for statistically significant predictors. As noted in the Model Specifications section, the use of deviation coding yields $k$-1 variables that test the difference between a given level of the categorical variable and the reference level. Region-based MdRQA models used the successful trials, monolingual language experience, middle tempi, and region 1 as reference categories; for more on mathematical interpretation of deviation-coded interaction terms, see Barr et al. (2013) [51].

| %REC | | | | | | |
|---|---|---|---|---|---|---|
| *Marginal R2 = 0.075, Conditional R2 = 0.084* | | | | | | |
| | Estimate | SE | df | t-value | p-value | ß |
| (Intercept) | 20.214 | 0.690 | 22.92 | 29.275 | < 0.0001 | – |
| Unsuccessful trials (v. Successful) | -7.930 | 1.188 | 518.07 | -6.676 | < 0.0001 | -.21 *** |
| Multilingual participants (v. Monolingual) | -0.022 | 1.381 | 22.92 | -0.016 | 0.987 | – |
| Region 2 (v. Region 1) | -2.098 | 1.397 | 1177.85 | -1.502 | 0.133 | -.08 ** |
| Region 3 (v. Region 1) | -2.000 | 1.397 | 1177.85 | -1.432 | 0.152 | – |
| Upper tempo range (v. Middle tempo range) | -0.554 | 1.428 | 1202.04 | -0.388 | 0.698 | – |
| Lower tempo range (v. Middle tempo range) | -2.215 | 1.253 | 1201.81 | -1.769 | 0.077 | -.05 . |
| Trial type x Language experience | -3.365 | 2.376 | 518.07 | -1.416 | 0.157 | – |
| Trial type x Upper tempo range | -2.530 | 2.874 | 1170.84 | -0.880 | 0.379 | – |
| Trial type x Lower tempo range | -6.468 | 2.519 | 1180.03 | -2.567 | 0.010 | -.08 ** |
| Language experience x Upper tempo | 2.626 | 2.857 | 1202.04 | 0.919 | 0.358 | – |
| Language experience x Lower tempo | 1.225 | 2.505 | 1201.81 | 0.489 | 0.625 | – |
| Trial type x Region 2 | -3.469 | 2.793 | 1177.85 | -1.242 | 0.215 | – |
| Trial type x Region 3 | 3.915 | 2.793 | 1177.85 | 1.402 | 0.161 | – |
| Language experience x Region 2 | -4.195 | 2.793 | 1177.85 | -1.502 | 0.133 | – |
| Language experience x Region 3 | -1.028 | 2.793 | 1177.85 | -0.368 | 0.713 | – |
| Upper tempo x Region 2 | -4.290 | 3.486 | 1177.85 | -1.231 | 0.219 | – |
| Lower tempo x Region 2 | -5.013 | 3.058 | 1177.85 | -1.639 | 0.101 | – |
| Upper tempo x Region 3 | -2.267 | 3.486 | 1177.85 | -0.650 | 0.516 | – |
| Lower tempo x Region 3 | 0.115 | 3.058 | 1177.85 | 0.037 | 0.970 | – |
| Trial type x Language exp. x Upper tempo | 1.970 | 5.749 | 1170.84 | 0.343 | 0.732 | – |
| Trial type x Language exp. x Lower tempo | 1.942 | 5.039 | 1180.03 | 0.385 | 0.700 | – |
| Trial type x Language exp. x Region 2 | -5.963 | 5.586 | 1177.85 | -1.068 | 0.286 | – |
| Trial type x Language exp. x Region 3 | 1.671 | 5.586 | 1177.85 | 0.299 | 0.765 | – |
| Trial type x Upper tempo x Region 2 | 0.258 | 6.972 | 1177.85 | 0.037 | 0.970 | – |
| Trial type x Lower tempo x Region 2 | 10.847 | 6.116 | 1177.85 | 1.774 | 0.076 | .06 . |
| Trial type x Upper tempo x Region 3 | 9.214 | 6.972 | 1177.85 | 1.322 | 0.187 | – |
| Trial type x Lower tempo x Region 3 | 7.666 | 6.116 | 1177.85 | 1.254 | 0.210 | – |
| Language exp. x Upper tempo x Region 2 | 8.824 | 6.972 | 1177.85 | 1.266 | 0.206 | – |
| Language exp. x Lower tempo x Region 2 | -5.888 | 6.116 | 1177.85 | -0.963 | 0.336 | – |
| Language exp. x Upper tempo x Region 3 | 8.148 | 6.972 | 1177.85 | 1.169 | 0.243 | – |
| Language exp. x Lower tempo x Region 3 | -9.221 | 6.116 | 1177.85 | -1.508 | 0.132 | – |
| Tr. type x Lang. exp. x Upper temp. x Reg. 2 | -7.662 | 13.945 | 1177.85 | -0.549 | 0.583 | – |
| Tr. type x Lang. exp. x Lower temp. x Reg. 2 | -4.407 | 12.232 | 1177.85 | -0.360 | 0.719 | – |
| Tr. type x Lang. exp. x Upper temp. x Reg. 3 | -2.761 | 13.945 | 1177.85 | -0.198 | 0.843 | – |
| Tr. type x Lang. exp. x Lower temp. x Reg. 3 | 1.518 | 12.232 | 1177.85 | 0.124 | 0.901 | – |
| %DET | | | | | | |
| *Marginal R2 = 0.098, Conditional R2 = 0.170* | | | | | | |
| | Estimate | SE | df | t-value | p-value | ß |
| (Intercept) | 14.248 | 1.628 | 23.53 | 8.753 | < 0.0001 | – |
| Unsuccessful trials (v. Successful) | -10.716 | 1.655 | 1117.69 | -6.473 | < 0.0001 | -.20 *** |
| Multilingual participants (v. Monolingual) | -0.805 | 3.256 | 23.53 | -0.247 | 0.807 | – |
| Region 2 (v. Region 1) | 1.916 | 1.828 | 1179.76 | 1.048 | 0.295 | – |

*(Continued)*

**Table 4.** (*Continued*)

| | | | | | | |
|---|---|---|---|---|---|---|
| Region 3 (v. Region 1) | -5.143 | 1.828 | 1179.76 | -2.813 | 0.005 | -.09 ** |
| Upper tempo range (v. Middle tempo range) | 0.434 | 1.881 | 1191.33 | 0.231 | 0.818 | – |
| Lower tempo range (v. Middle tempo range) | -3.894 | 1.650 | 1192.66 | -2.360 | 0.018 | -.07 * |
| Trial type x Language experience | -0.857 | 3.311 | 1117.69 | -0.259 | 0.796 | – |
| Trial type x Upper tempo range | 1.534 | 3.821 | 1201.50 | 0.402 | 0.688 | – |
| Trial type x Lower tempo range | 1.412 | 3.344 | 1200.28 | 0.422 | 0.673 | – |
| Language experience x Upper tempo | 1.301 | 3.762 | 1191.33 | 0.346 | 0.730 | – |
| Language experience x Lower tempo | -1.861 | 3.300 | 1192.66 | -0.564 | 0.573 | – |
| Trial type x Region 2 | 1.117 | 3.656 | 1179.76 | 0.305 | 0.760 | – |
| Trial type x Region 3 | 9.870 | 3.656 | 1179.76 | 2.700 | 0.007 | .09 ** |
| Language experience x Region 2 | -7.312 | 3.656 | 1179.76 | -2.000 | 0.046 | -.06 * |
| Language experience x Region 3 | 4.158 | 3.656 | 1179.76 | 1.137 | 0.256 | – |
| Upper tempo x Region 2 | -2.983 | 4.563 | 1179.76 | -0.654 | 0.513 | – |
| Lower tempo x Region 2 | -4.659 | 4.003 | 1179.76 | -1.164 | 0.245 | – |
| Upper tempo x Region 3 | 4.009 | 4.563 | 1179.76 | 0.878 | 0.380 | – |
| Lower tempo x Region 3 | 2.559 | 4.003 | 1179.76 | 0.639 | 0.523 | – |
| Trial type x Language exp. x Upper tempo | 8.666 | 7.642 | 1201.50 | 1.134 | 0.257 | – |
| Trial type x Language exp. x Lower tempo | -0.060 | 6.688 | 1200.28 | -0.009 | 0.993 | – |
| Trial type x Language exp. x Region 2 | -3.186 | 7.312 | 1179.76 | -0.436 | 0.663 | – |
| Trial type x Language exp. x Region 3 | 5.464 | 7.312 | 1179.76 | 0.747 | 0.455 | – |
| Trial type x Upper tempo x Region 2 | -5.961 | 9.127 | 1179.76 | -0.653 | 0.514 | – |
| Trial type x Lower tempo x Region 2 | 3.349 | 8.005 | 1179.76 | 0.418 | 0.676 | – |
| Trial type x Upper tempo x Region 3 | 16.144 | 9.127 | 1179.76 | 1.769 | 0.077 | .06 . |
| Trial type x Lower tempo x Region 3 | -1.614 | 8.005 | 1179.76 | -0.202 | 0.840 | – |
| Language exp. x Upper tempo x Region 2 | 10.850 | 9.127 | 1179.76 | 1.189 | 0.235 | – |
| Language exp. x Lower tempo x Region 2 | -8.186 | 8.005 | 1179.76 | -1.023 | 0.307 | – |
| Language exp. x Upper tempo x Region 3 | 28.537 | 9.127 | 1179.76 | 3.127 | 0.002 | .11 ** |
| Language exp. x Lower tempo x Region 3 | -6.732 | 8.005 | 1179.76 | -0.841 | 0.401 | – |
| Tr. type x Lang. exp. x Upper temp. x Reg. 2 | -17.592 | 18.253 | 1179.76 | -0.964 | 0.335 | – |
| Tr. type x Lang. exp. x Lower temp. x Reg. 2 | 5.424 | 16.011 | 1179.76 | 0.339 | 0.735 | – |
| Tr. type x Lang. exp. x Upper temp. x Reg. 3 | 6.188 | 18.253 | 1179.76 | 0.339 | 0.735 | – |
| Tr. type x Lang. exp. x Lower temp. x Reg. 3 | 6.243 | 16.011 | 1179.76 | 0.390 | 0.697 | – |

**MAXL**
*Marginal R2 = 0.192, Conditional R2 = 0.324*

| | Estimate | SE | df | t-value | p-value | ß |
|---|---|---|---|---|---|---|
| (Intercept) | 1.013 | 0.084 | 23.30 | 12.022 | < 0.0001 | – |
| Unsuccessful trials (v. Successful) | -0.698 | 0.062 | 1196.00 | -11.284 | < 0.0001 | -.32 *** |
| Multilingual participants (v. Monolingual) | -0.147 | 0.169 | 23.30 | -0.869 | 0.394 | – |
| Region 2 (v. Region 1) | -0.028 | 0.067 | 1180.00 | -0.412 | 0.680 | – |
| Region 3 (v. Region 1) | -0.363 | 0.067 | 1180.00 | -5.393 | < 0.0001 | -.16 *** |
| Upper tempo range (v. Middle tempo range) | -0.023 | 0.069 | 1186.00 | -0.331 | 0.741 | – |
| Lower tempo range (v. Middle tempo range) | -0.247 | 0.061 | 1188.00 | -4.053 | < 0.0001 | -.11 *** |
| Trial type x Language experience | 0.068 | 0.124 | 1196.00 | 0.554 | 0.580 | – |
| Trial type x Upper tempo range | -0.065 | 0.141 | 1194.00 | -0.461 | 0.645 | – |
| Trial type x Lower tempo range | -0.003 | 0.124 | 1193.00 | -0.021 | 0.983 | – |
| Language experience x Upper tempo | 0.043 | 0.139 | 1186.00 | 0.313 | 0.754 | – |
| Language experience x Lower tempo | 0.007 | 0.122 | 1188.00 | 0.059 | 0.953 | – |
| Trial type x Region 2 | -0.280 | 0.135 | 1180.00 | -2.083 | 0.037 | -.06 * |
| Trial type x Region 3 | 0.446 | 0.135 | 1180.00 | 3.311 | 0.001 | .10 *** |

(*Continued*)

**Table 4.** (Continued)

| | | | | | | |
|---|---|---|---|---|---|---|
| Language experience x Region 2 | -0.323 | 0.135 | 1180.00 | -2.402 | 0.016 | -.07 * |
| Language experience x Region 3 | 0.085 | 0.135 | 1180.00 | 0.632 | 0.527 | – |
| Upper tempo x Region 2 | -0.257 | 0.168 | 1180.00 | -1.531 | 0.126 | – |
| Lower tempo x Region 2 | -0.097 | 0.147 | 1180.00 | -0.659 | 0.510 | – |
| Upper tempo x Region 3 | -0.221 | 0.168 | 1180.00 | -1.313 | 0.189 | – |
| Lower tempo x Region 3 | 0.233 | 0.147 | 1180.00 | 1.579 | 0.115 | – |
| Trial type x Language exp. x Upper tempo | 0.333 | 0.282 | 1194.00 | 1.181 | 0.238 | – |
| Trial type x Language exp. x Lower tempo | 0.133 | 0.247 | 1193.00 | 0.539 | 0.590 | – |
| Trial type x Language exp. x Region 2 | -0.145 | 0.269 | 1180.00 | -0.539 | 0.590 | – |
| Trial type x Language exp. x Region 3 | -0.192 | 0.269 | 1180.00 | -0.714 | 0.476 | – |
| Trial type x Upper tempo x Region 2 | 0.033 | 0.336 | 1180.00 | 0.097 | 0.923 | – |
| Trial type x Lower tempo x Region 2 | 0.345 | 0.295 | 1180.00 | 1.171 | 0.242 | – |
| Trial type x Upper tempo x Region 3 | 0.557 | 0.336 | 1180.00 | 1.658 | 0.098 | .05 . |
| Trial type x Lower tempo x Region 3 | -0.143 | 0.295 | 1180.00 | -0.486 | 0.627 | – |
| Language exp. x Upper tempo x Region 2 | 0.574 | 0.336 | 1180.00 | 1.709 | 0.088 | .05 . |
| Language exp. x Lower tempo x Region 2 | -0.148 | 0.295 | 1180.00 | -0.504 | 0.615 | – |
| Language exp. x Upper tempo x Region 3 | 0.943 | 0.336 | 1180.00 | 2.805 | 0.005 | .09 *** |
| Language exp. x Lower tempo x Region 3 | -0.295 | 0.295 | 1180.00 | -1.002 | 0.316 | – |
| Tr. type x Lang. exp. x Upper temp. x Reg. 2 | -0.238 | 0.672 | 1180.00 | -0.354 | 0.723 | – |
| Tr. type x Lang. exp. x Lower temp. x Reg. 2 | 0.115 | 0.590 | 1180.00 | 0.195 | 0.845 | – |
| Tr. type x Lang. exp. x Upper temp. x Reg. 3 | -0.438 | 0.672 | 1180.00 | -0.652 | 0.514 | – |
| Tr. type x Lang. exp. x Lower temp. x Reg. 3 | 0.635 | 0.590 | 1180.00 | 1.077 | 0.282 | – |

The interaction between trial type and region on MAXL revealed findings similar to those for %DET. MAXL also peaked during region 2 for Unsuccessful Trials, which supports H2. However, MAXL peaked during region 1 for Successful Trials, opposing H2. Both trial types produced the smallest MAXL values during region 3. Together, these results suggested that those who could not successfully phase experienced the greatest attractor strength during anti-phase, while those who successfully phased experienced the greatest attractor strength during their initial desynchronization from the metronome.

Interestingly, regardless of level of success with the phasing task, participants experienced the weakest coupling with the metronome when moving from antiphase to in-phase synchrony. To the contrary, human-metronome coupling was relatively stronger when shifting from synchrony to antiphase. This was demonstrated by the relative stability of taps (i.e., increased number of consecutive recurrent states) in region 1 compared to region 3, indicating differential pulls of in- and anti-phase attractors. Since in-phase synchrony is known to be a stronger attractor than antiphase, it may be more difficult to escape from synchrony toward antiphase; thus, participants passed more slowly through region 1 than through region 3. Participants resynchronized more quickly, possibly because they were moving from the weaker attractor of antiphase toward the stronger attractor of in-phase synchrony.

The significant interactions among language experience, tempo, and region on both %DET (Fig 5B) and MAXL (Fig 5C) partially supported H2. We expected participants to reach peak %DET and MAXL values during region 2 due to increased stability near antiphase. This held true across tempi for monolingual speakers, but multilinguals exhibited more variability across tempi and regions. This result suggested that tapping predictability and attractor strength varied with task parameters. The relationship between selective attention and pull toward the metronome at various relative phases is expanded upon during our discussion of H3.

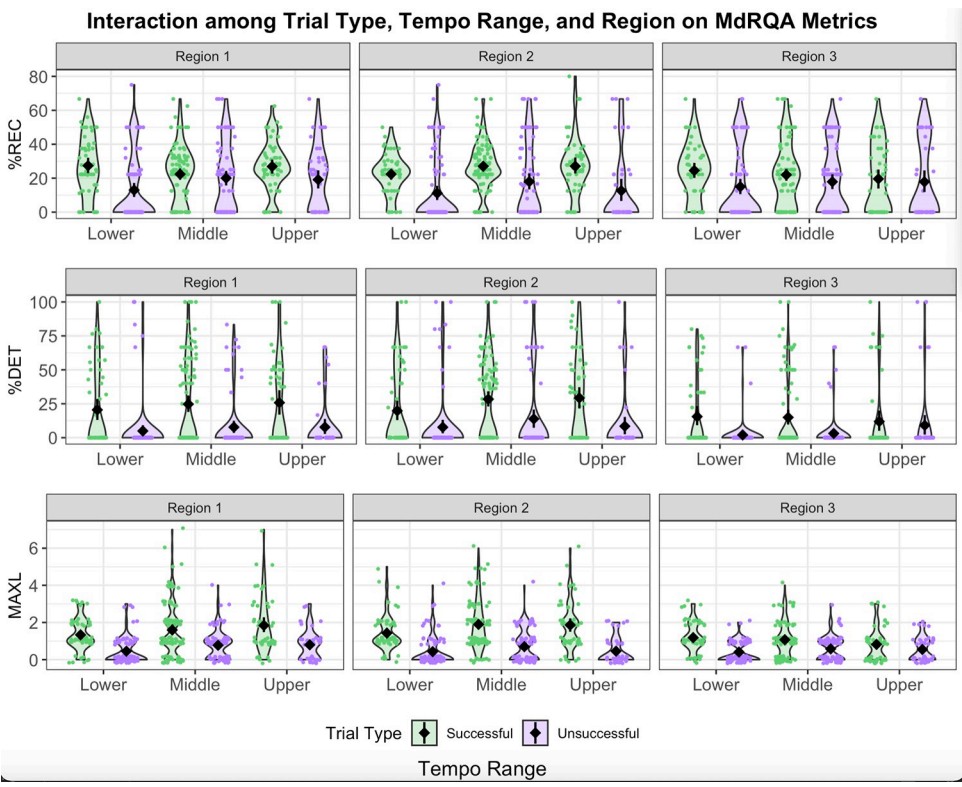

**Fig 4. Effects of trial type (green: Successful; purple: Unsuccessful), tempo range, and region on MdRQA metrics: %REC (Panel A, top), %DET (Panel B, middle), and MAXL (Panel C, bottom).** The metronome ranged from 80–140 bpm. In our linear mixed effects model, we binned tempo into the following categories: lower (80–90 bpm), middle (100–120 bpm), and upper (130–140 bpm). The violin plots depict the probability density of the binned tempo data at different %REC, %DET, and MAXL values. The mean value is indicated by a diamond near the center of each violin. Results of statistical analyses are located in Table 4.

## Hypothesis 3

In H3, we predicted that monolingual speakers would experience stronger coupling with the metronome and therefore produce greater %REC, %DET, and MAXL than multilingual speakers would. In other words, because multilingual speakers have been shown to have greater inhibitory control compared with monolinguals, we hypothesized that multilingual participants would be better able to intentionally decouple from the metronome in order to successfully phase, as compared to monolingual participants' anticipated difficulty overcoming the pull toward in- and antiphase tapping. We use general MdRQA to assess the effects of language experience irrespective of phasing region, and then we use region-based MdRQA to compare how monolinguals and multilinguals differed during specific regions (i.e., regions 1–3). The significant outcomes related to hypothesis 3 include an interaction effect between language experience and tempo on %REC for general MdRQA (Fig 3A), as well as interaction effects among language experience, tempo, and region on both %DET (Fig 5B) and MAXL (Fig 5C) for region-based MdRQA. Results of the statistical analyses for H3 are available in Tables 2 and 4.

As measured by %REC, tapping data became less noisy as tempo increased. Multilinguals demonstrated a sharper and greater increase than monolinguals. These findings failed to support H3, perhaps indicating that %REC during intentional decoupling is not tied to inhibitory control—a connection that had been the foundation for H3.

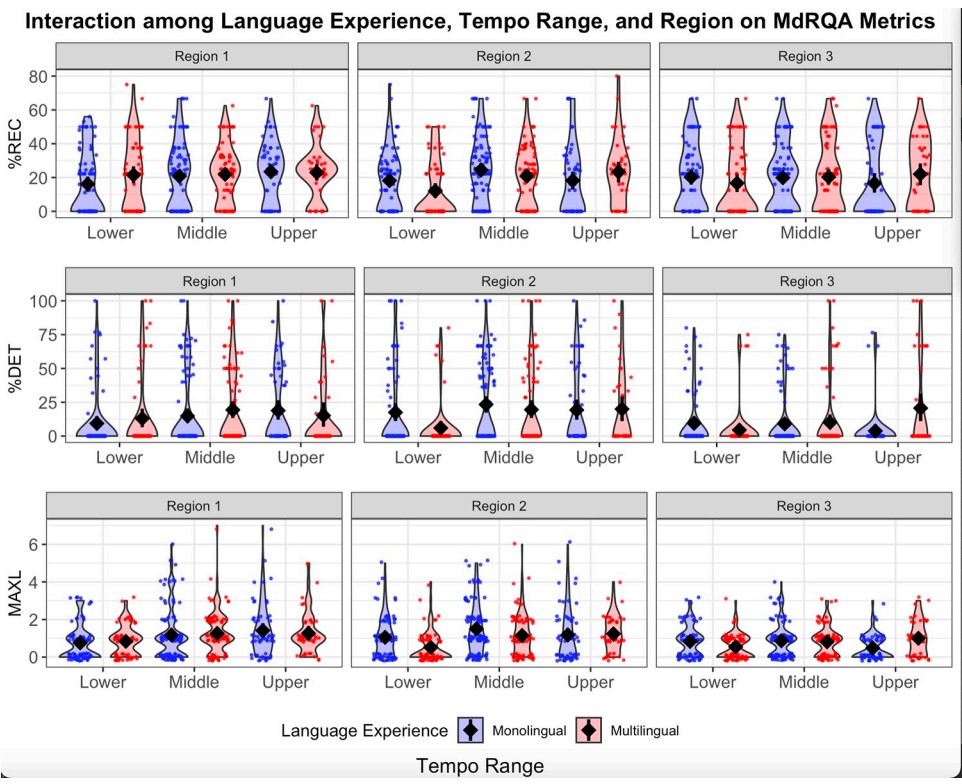

**Fig 5. Effects of language experience (red: Multilingual; blue: Monolingual), tempo range, and region on MdRQA metrics: %REC (Panel A, top), %DET (Panel B, middle), and MAXL (Panel C, bottom).** The metronome ranged from 80–140 bpm. In our linear mixed effects model, we binned tempo into the following categories: lower (80–90 bpm), middle (100–120 bpm), and upper (130–140 bpm). The violin plots depict the probability density of the binned tempo data at different %REC, %DET, and MAXL values. The mean value is indicated by a diamond near the center of each violin. Results of statistical analyses are located in Table 4.

The significant interactions among language experience, tempo, and region on %DET and MAXL showed a pattern of results similar to those of %REC. Both %DET and MAXL increased with tempo across regions 1, 2, and 3 for multilingual speakers. Thus, the predictability (indexed by %DET) and attractor strength (indexed by MAXL) of multilinguals' taps grew as tempo increased during all regions. Again, in contrast with H3, these findings suggested that multilingual speakers had more difficulty desynchronizing and resynchronizing with the metronome at faster tempi.

Monolingual speakers exhibited a more complex pattern: %DET and MAXL increased with tempo during region 1, remained stable across tempi during region 2, and decreased with tempo during region 3. This meant that—for monolinguals—the predictability and attractor strength of the human-metronome system were greater at faster tempi when desynchronizing and greater at slower tempi when resynchronizing. Monolingual participants had more difficulty decoupling from synchrony at faster tempi and more difficulty returning to synchrony at slower tempi.

Overall, these interactions for %DET and MAXL neither fully supported nor contradicted our predictions regarding language experience, providing a complex picture of the impacts of intentional decoupling across task complexity. Task parameters affected monolingual and multilingual participants differently. For example, multilinguals and monolinguals had opposite relationships with tempo during region 3. Suggestions for how to disentangle these findings are provided in Limitations and Future Directions.

### *Post hoc* analyses

We did not hypothesize about how success in the phasing task would be reflected by MdRQA metrics, as we did not anticipate such a high percentage of Unsuccessful Trials. As such, we conducted exploratory analyses to identify how Successful and Unsuccessful Trials differed in their dynamics.

*Post hoc* analyses revealed that Successful and Unsuccessful trials exhibited significantly different dynamics and metrics (see Fig 4 and Table 4). %REC, %DET, and MAXL are all significantly greater for Successful Trials than for Unsuccessful Trials. Furthermore, Successful Trials generally supported our prediction that region 2 should yield the most structured tapping data because of the antiphase attractor. %REC, %DET, and MAXL peak during region 2 and are the lowest during region 3 for Successful Trials. Unsuccessful Trials showed a different pattern of results: Region 2 was the least noisy (as indicated by %REC) and most structured (as indicated by %DET), but attractor strength (as indicated by MAXL) waned from region 1 to region 3. The absence of a clear pattern demonstrated by Unsuccessful Trials supports the interpretation that these trials were characterized by substantially different dynamics than Successful Trials.

While many participants faced difficulty phasing, the dynamics of Successful Trials generally replicated the dynamics identified in Kim's [52] analysis, in which expert percussionists in Schutz's case study [4] were found to dwell near in-phase and antiphase attractors but to move between them rather quickly. One notable difference in our results is the absence of quick transitions from initial synchrony to antiphase (demonstrated by relatively high metrics for region 1); however, this difference may have resulted from the difference between phasing with an adaptive human partner versus a rigid metronome. Similarities between the Successful Trials in the current work and Schutz's study, however, held across music experience, tempo, and task demands, suggesting that perception-action coordination dynamics during successful phasing may emerge from general principles of motor behavior and intentional decoupling.

### Limitations and future directions

The present study compared Schutz's [4] expert study to a broader population of participants using a simplified version of the original task. We replicated the phasing dynamics in non-experts during Successful Trials. Participants in Unsuccessful Trials were unable to detect $\psi$, meaning they failed to complete one round of phasing; these trials did not replicate patterns in Schutz's case study. This raises the question of what shapes phasing ability. Previous literature suggests attentional flexibility [53] and neuromuscular-skeletal constraints [54] predict high-level motor skill. Although we were able to identify distinct dynamics, our study did not permit us to investigate potential reasons for differences. Future work should explore the constraints that shape the distribution of Incomplete and Unsuccessful Trials relative to Successful Trials.

While phasing dynamics of Successful Trials were similar to those observed between expert musicians [4], we do not claim that nonmusicians and non-professional musicians should be identical to expert musicians in their phasing abilities: Critical differences in the dynamics may emerge when we examine interactive phasing context. To that end, future research should utilize a dyadic phasing task with a wider tempo range and again evaluate musical and linguistic experience to investigate whether our observations extend to dyadic conditions.

Finally, future work should develop a dynamical model that captures the observed behavior and provides novel testable predictions. Such models have been influential in the study of coordination dynamics [55]. A model of phasing should account for the observed changes in attractor landscape determined by task demands and individual experience from the present work and the original case study [4] and should provide an account of dyadic phasing.

## Conclusion

Inspired by Schutz's [4] data-driven case study, we here introduced a novel phasing task that requires intentional decoupling from an auditory metronome. Our complementary approach allowed us to study perception-action coordination through traditional in-phase and antiphase tapping paradigms: Despite key differences between the two paradigms (i.e., partnered expert performance versus isochronous human-metronome phasing), we conceptually replicated the original findings [4]—that is, dwelling near in-phase and antiphase and quickly transitioning between these attractors—among participants who were able to successfully phase, regardless of individual experiences or task demands. Given these findings, similar sensorimotor coordination processes may underlie successful phasing, even for different populations performing at different rhythmic complexities (e.g., isochronous versus non-isochronous). Parallel findings from two very different populations completing similar tasks of varying difficulty provides converging evidence about the general dynamics of phasing and perception-action coordination.

## Supporting information

**S1 Appendix. Descriptions of practice sessions.**
(DOCX)

**S2 Appendix.**
(DOCX)

**S3 Appendix.**
(DOCX)

## Acknowledgments

The authors would like to thank Edward Large (University of Connecticut) for advice, feedback, and guidance and Matthew B. Jané (University of Connecticut) for his assistance designing the figures. This work was completed in part as coursework done by lead author C. Hall in the "Applications of Nonlinear Time Series Analyses" graduate course at the University of Connecticut, taught by coauthor A. Paxton and Steven J. Harrison.

## Author Contributions

**Conceptualization:** Ji Chul Kim.

**Data curation:** Caitrín Hall, Ji Chul Kim.

**Formal analysis:** Caitrín Hall, Ji Chul Kim, Alexandra Paxton.

**Funding acquisition:** Caitrín Hall.

**Investigation:** Caitrín Hall, Ji Chul Kim.

**Methodology:** Caitrín Hall, Ji Chul Kim, Alexandra Paxton.

**Project administration:** Caitrín Hall, Ji Chul Kim.

**Software:** Caitrín Hall, Ji Chul Kim.

**Supervision:** Ji Chul Kim, Alexandra Paxton.

**Visualization:** Ji Chul Kim, Alexandra Paxton.

**Writing – original draft:** Caitrín Hall.

**Writing – review & editing:** Caitrín Hall, Ji Chul Kim, Alexandra Paxton.

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
