## [Decision Letter · Decision Letter 0]

24 Nov 2021

PONE-D-21-26808Multidimensional Recurrence Quantification Analysis of Human-Metronome PhasingPLOS ONE

Dear Dr. Hall,

Thank you for submitting your manuscript to PLOS ONE. I apologize for the long delay, but it took a while to find two expert reviewers that had time to closely read your manuscript. Unfortunately, the reviewers do not quite agree on the magnitude of changes that are needed for your manuscript to be accepted for publication. However, weighting all things, I decided to invite you to submit a revised version of the manuscript that addresses the points raised during the review process. You will see that both reviews agree that more information needs to be provided regarding the particular analysis, parameters settings, and presentation/interpretation of the results. However, reviewer 2 is generally more critical, and also lists a whole series of shortcomings of the paper that concern its conceptual nature, but also the general way that various aspects of literature, data, design and analysis are communicated If you consider submitting a revision of your paper, please make sure that you address all concerns raised by the two reviewers point-by-point. Particularly, it will be important that you can convice reviewer 2 that your revised version has taken majors steps towards a general improvement over the current mansucript.

We look forward to receiving your revised manuscript.

Kind regards,

Sebastian Wallot, Ph.D

Academic Editor

PLOS ONE

2. Please change "female” or "male" to "woman” or "man" as appropriate, when used as a noun (see for instance https://apastyle.apa.org/style-grammar-guidelines/bias-free-language/gender).

3. Thank you for stating the following in the Competing Interests/Financial Disclosure:

I have read the journal's policy and the authors of this manuscript have the following competing interests: EWL holds ownership interest in Oscilloscape, LLC.

We note that one or more of the authors are employed by a commercial company: EWL holds ownership interest in Oscilloscape, LLC.

Within your Competing Interests Statement, please confirm that this commercial affiliation does not alter your adherence to all PLOS ONE policies on sharing data and materials by including the following statement: ""This does not alter our adherence to  PLOS ONE policies on sharing data and materials.” (as detailed online in our guide for authors http://journals.plos.org/plosone/s/competing-interests) . If this adherence statement is not accurate and  there are restrictions on sharing of data and/or materials, please state these. Please note that we cannot proceed with consideration of your article until this information has been declared.

4. We note that Figure 2 in your submission contain copyrighted images. All PLOS content is published under the Creative Commons Attribution License (CC BY 4.0), which means that the manuscript, images, and Supporting Information files will be freely available online, and any third party is permitted to access, download, copy, distribute, and use these materials in any way, even commercially, with proper attribution. For more information, see our copyright guidelines: http://journals.plos.org/plosone/s/licenses-and-copyright.

5. Please include a caption for figure 8 in manuscript.

Reviewers' comments:

Reviewer's Responses to Questions

**Comments to the Author**

1. Is the manuscript technically sound, and do the data support the conclusions?

Reviewer #1: Partly

Reviewer #2: No

2. Has the statistical analysis been performed appropriately and rigorously? 

Reviewer #1: Yes

Reviewer #2: No

3. Have the authors made all data underlying the findings in their manuscript fully available?

Reviewer #1: Yes

Reviewer #2: Yes

4. Is the manuscript presented in an intelligible fashion and written in standard English?

Reviewer #1: Yes

Reviewer #2: Yes

5. Review Comments to the Author

Reviewer #1: The current study aims to test the roles of multilingualism and tempo in predicting perception-action coordination. To evaluate these effects, the authors conducted a phasing-based experiment – tapping a tempo deviated from a specific BPM played by a metronome. Innovatively, the sample did not consist of professional musicians. Due to better selective attention, the researchers hypothesized that multilingual participants would show higher rates of success in the task – defined as a return to in-phase tap after one lap of desynchronization. Further, they assessed that around both antiphase and in-phase subjects would tap more consistently, and that 100-120 BPM would be the most compatible tempi to succeed in phasing. The reasoning to hypothesize the latter was vague.

Multi-dimensional Recurrence Quantification Analysis was applied on components of the relative phase between participants' taps and original tempo set by the metronome. In this manner, the researchers evaluated several aspects of the performance on the task. Their findings did not support the hypothesis regarding the suitable tempo for phasing. Instead, they found that participants are drawn more to a specific relative phase as the tempo increases from 80 to 140 BPM. Windowed mdRQA implied that in-phase is a stronger attractor than antiphase on trials with higher tempi. While monolinguals transition in and out of synchrony is affected by tempo, multilingual subjects could keep a stable tapping rhythm across the range of BPM – corresponding with the authors' hypothesis. However, a variance test was not provided.

The paradigm was based on previous research and fitted to non-musicians, hence helping to generalize Schutz's previous findings regarding phasing. The authors declared their hypotheses precisely and returned to them in the discussion section. Also, Recurrence Quantification Analysis was an innovative and suitable way to approach this data – supported by the researchers' findings of higher mdRQA attributes on successful trials. Moreover, the supplemented data, as well as the videos, and codes were plain and explanatory.

My primary concerns are:

• The reasoning behind conducting mdRQA on the sine and cosine of the relative phase is unclear. How should a recurrence point of cosine and sine be interpreted?

• The windowed analysis’s relatively high SD for the mdRQA outcomes (REC, DET, maxL) suggests that the time series may be too short to conduct mdRQA properly. To address this point, the authors should check the average size of the recurrence plots and the histogram of the mdRQA outcomes.

• The researchers should provide the mdRQA parameters that were used for the analysis.

My minor concerns are:

• Effect size would be an essential addition to the Tukey posthoc, specifically when the degrees of freedom are high.

• In my opinion, a recurrence point should not be defined as a repetition of the same value. In this case, it seems that a recurrence point stands for similarity dependent on the chosen radius.

• Why does mdRQA fit this data better than CRQA?

If the authors address the above queries, I think the paper could contribute to the literature.

Reviewer #2: Title:

Multidimensional Recurrence Quantification Analysis of Human-Metronome Phasing

Comments to the editor and authors:

I would like to thank the authors for the opportunity to read this manuscript. The paper reports an empirical study of a ‘phasing’ task using multidimensional recurrence quantification analysis. While I appreciate the effort put in this work, the manuscript has major problems and the overall quality is far below adequate for publication. In brief, the place of this study within past research is not well established, the task is not described in enough detail, the analyses (MdRQA, regression, and posthoc tests) are not presented in enough detail, the results are not properly reported and interpreted, and some of the analyses reported are not appropriate to assess the hypotheses put forward. In addition, the text is unclear in many passages, and the limitations are not addressed.

Therefore, I have no alternative than to recommend rejecting this submission.

It would be impractical to list all problems I found in this manuscript. Below I will list some of the main issues. I hope the authors will see my comments are meant to help them improve the manuscript for a future submission.

L75ff: about RQA

The analytical framework of RQA is presented very superficially as a method “to describe nonlinearities of coupled dynamical systems”. The reader must be presented with a bit more of the conceptual (if not mathematical) background.

L76ff: about MdRQA.

Wallot et al (2016) present MdRQA as a method to “analyze group-level behavior of groups bigger than a dyad.” Given the current study design can be conceived of as involving human-metronome 'dyads', you must explain better why you chose MdRQA. Further below in the text you indicate that using MdRQA was not the initial choice but an ad hoc decision, a workaround because of issues with the data. This must be made clear.

L72: “The “static” part actually varies along with the moving part…”

This is too vague. What is it that you want the reader to attend?

L85:

Calling 25 WEIRD undergraduates with age range 18-21 “a broad population… with different musical and linguistic backgrounds” is not realistic.

L92ff Hypotheses

1) Please indicate your hypotheses and the predictions tested more clearly. I identified 3 (or perhaps 4) hypotheses/predictions:

H1: “participants will demonstrate stable tapping near in-phase and antiphase” with prediction “higher metrics during those periods.”

To test H1, I suppose one would compare inphase and antiphase, which seems related to the windowed analysis, but this is unclear. The motivation to conduct windowed MdRQA is said to be “to compare successful and unsuccessful trials” so the analytical framework and the implications for the hypothesis are confusing.

H2A: “multilingual participants may be able to simultaneously attend to the metronome while adopting a different tempo.” H2B: “Monolingual participants may experience stronger coupling with the metronome” with prediction: “greater metrics for monolingual speakers.”

To test H2, the idea seems to be to use regression analysis, but since the interactions are not properly taken into account in the report, you end up not able to compare between monolingual and multilingual participants appropriately.

Also, given you simplified the task, the prediction in L102 does not seem well justified.

H3: “we expect the middle range of our selected tempi (100-120 bpm) to yield the most structured phasing performance for all participants” with prediction “higher metrics”

To test H3, I suppose one would compare middle range with lower (80-100) and higher (120-140) ranges. But in this case, adding ‘tempo’ as a linear predictor in the models does not seem to be the appropriate way to test this. Because your prediction is that the relationship is not linear but perhaps something like parabolic: low-high-low

2) How does comparing successful and unsuccessful trials relate to the hypotheses presented? In other words, why did you add trial type as a predictor in the models?

L112: sample size

What you report is not what is commonly meant by 'determining sample size'. What exactly was the sample size determined in advance? Anything from 15 – 30? Which 'groups' did you consider relevant to determine sample size? Considering the vars gender (male/female), linguistic abilities (mono/multi), and musical experience (yes/no), you do not have 15-30 in each group.

L115 Data inclusion criteria and n of trials

Given you need to know what a trial is to understand this part, I suggest you move it to after you describe the procedure, i.e. before Data Analysis.

L115

Please justify the number of trials in the dataset. What counts as a trial, how many trials per participant, how many issues e.g. due to early termination, malfunction?

L116 , then L137

Please clarify what you mean by ‘one round of phasing’ and “complete one phasing lap”. The whole procedure is a bit unclear for the reader

L119

Given the instructions to participants, how can you not have any tapping data in some trials? What is the justification to recode these missing data as zero? If this was done merely to be able to fit the models, this is probably wrong.

L131

You mention demographics survey and then do not report/discuss anything. What exactly do you mean here?

L141ff Data analysis

1) You must report the parameters used in MdRQA and how you decided which values to use.

2) The Statistical analyses conducted must be properly described. Simply saying that you calculated descriptive and inferential statistics is too vague. You fitted regression models and you must describe what you did in detail so that readers can understand and assess your work. The structure of the models fitted must be clearly explained. For example, the tables suggests you included interactions and you did not mention that. Describe each predictor, their possible values and scale. Were they centered or standardized prior to including in the models? How did you account for repeated measures arising from multiple trials? The [ ] notation in the table suggests you used 'random' effects, if so what and why and how? One table indicates you added a 4-way interaction: trialtype*language*tempo*window but not all parameter combinations seem present. As the moment it is not at all clear that the regression models were structured (declared) properly and I have very low confidence in the reported results and their interpretation. Also, Table 5 reports post hoc tests but you did not clearly explain them.

3) As it is, the text is saying you used all these R packages to calculate statistics, which is not the case (e.g. viridis). If you indeed want to acknowledge all packages you used, you must clarify their role in the analyses.

L146

Please add at least a sentence to clarify to the reader what phi is conceptually (relative phase between participants’ taps and metronome ticks). Please format the formula correctly to improve readability.

L154:

What do you mean by 'the apparent discontinuity in Ψ near synchrony'?

L154

1)The procedure to ‘adapt’ the data for MdRQA must be better motivated and better explained (to say you “decomposed each tap into its x- and y-coordinates by taking cosine and sine” is not very clear).

2)In what ways does the procedure adapt the data to mdrqa? How did this solve the problem of 'apparent discontinuity'?

L162 windowed analysis:

1)please clarify the motivation to conduct windowed analysis: is this related to H1? How does this relate to comparing successful and unsuccessful trials?

2)Did you identify the transitions from one window to another for each time series? How?

3)Fig 2: The figure does not show MdRQA but a MdR plot. Why is the in-phase period not represented in this MdRplot?

4) Fig 2: caption is incorrect. The time series may be perhaps 'represented' in both axes, but this plot shows recurrences not the time series. The axes labels need adjusting as does the title.

L177:

Certainly not “all results”, so please clarify which results do you report (e.g. summary statistics, table of estimates from regression models). You should also consider reporting some effect sizes (e.g. relevant slopes) in the text if you want to use them to support your inferences.

L184 Fig 3

Table 2 suggests the model included interactions. But Fig 3 shows raw data and does not take interactions into account. Also, what does the error bar represent?

A general note: The tables are poorly formatted and take too much space.

L213 “significant interaction for tempo, window, and trial type”

1) Please clarify this passage, as it is not simply 'the interaction' that is relevant here but the estimated effects. Also, it is incorrect to say interaction ‘for’ A, B, and C in this context.

2) in what ways did the results “conceptually replicate” previous findings?

L214

what do you mean by ‘relative stability’?

L229: “our finding that inhibitory control varies based on language experience and tempo”

Which results specifically do you think supports your claim, please report the relevant estimates, effect sizes, and statistics.

L289 Conclusion

It is not clear that your results provided a 'broad replication of the original dynamics' reported by Schultz.

L290 “sensorimotor coordination processes may underlie successful phasing”

what would it mean to say that sensorimotor coordination processes did NOT underlie successful phasing?

A general comment about the figures

Please indicate clearly what the error bars show. The titles of figures must be revised to correct typos, grammar mistakes and to improve their meaning, especially figures 2,4, 5, and 6. For ex, it makes no sense to say ‘interaction between language experience, window, and tempo on DET’

A general comment about openly available data

Please consider making the data available in a more friendly format such as a csv or txt, as matlab is perhaps too restrictive.

6. PLOS authors have the option to publish the peer review history of their article (what does this mean?). If published, this will include your full peer review and any attached files.

Reviewer #1: No

Reviewer #2: No

---

## [Author Response · Author response to Decision Letter 0]

6 Jul 2022

Academic editor Sebastian Wallot advised us to upload our response to the requested changes as a separate file. See "PLOS One Response to Reviewers."

---

## [Decision Letter · Decision Letter 1]

4 Aug 2022

PONE-D-21-26808R1Multidimensional recurrence quantification analysis of human-metronome phasingPLOS ONE

Dear Dr. Hall,

Thank you for submitting your manuscript to PLOS ONE. After careful consideration, we feel that it has merit but does not fully meet PLOS ONE’s publication criteria as it currently stands. Therefore, we invite you to submit a revised version of the manuscript that addresses the points raised during the review process. Both reviewers and I agree, that the manuscript has improved substantially. While reviewer 1 is satisfied with the changes you made, reviewer 2 has raised multiple concerns regarding how the inferential statistics are done and how some of the findings are interpreted with regard to possible inferences drawn from the models you ran, and also regarding the background of the hypotheses you formulated. In a second revision, please clarify these issues or adjust your modelling procedures accordingly.

We look forward to receiving your revised manuscript.

Kind regards,

Sebastian Wallot, Ph.D

Academic Editor

PLOS ONE

Journal Requirements:

Reviewers' comments:

Reviewer's Responses to Questions

**Comments to the Author**

1. If the authors have adequately addressed your comments raised in a previous round of review and you feel that this manuscript is now acceptable for publication, you may indicate that here to bypass the “Comments to the Author” section, enter your conflict of interest statement in the “Confidential to Editor” section, and submit your "Accept" recommendation.

Reviewer #1: All comments have been addressed

Reviewer #2: (No Response)

2. Is the manuscript technically sound, and do the data support the conclusions?

Reviewer #1: Yes

Reviewer #2: No

3. Has the statistical analysis been performed appropriately and rigorously? 

Reviewer #1: Yes

Reviewer #2: No

4. Have the authors made all data underlying the findings in their manuscript fully available?

Reviewer #1: Yes

Reviewer #2: Yes

5. Is the manuscript presented in an intelligible fashion and written in standard English?

Reviewer #1: Yes

Reviewer #2: Yes

6. Review Comments to the Author

Reviewer #1: The authors improved the paper and responded thoroughly to the points of concern. In my opinion, the manuscript can now be accepted.

Reviewer #2: My recommendation is major revision.

Thank you for the opportunity to revisit this manuscript. I appreciate the effort put by the authors in rewriting the manuscript. It has improved. For example, the hypotheses and procedures are clearer now. I can say that I understand the study better now. However, the manuscript still contains many major problems, and now that I understand the procedure and analyses a bit better, new problems became evident. Most importantly, the regression models fitted are not described in enough clarity (and at least one variable may be wrongly coded), many claims are made based on wrong interpretations of 'interactions' and/or with no statistical support (some examples below).

Additionally, I now see that what the authors call a 'windowed analysis' does not correspond to what the term means by other authors using RQA because they did not use overlapping windows. What the authors did was to divide the tapping time series into periods (which they call windows 1, 2 and 3) and use these periods ('windows') as predictors in the regression models to see if RQA measures wary across periods.

I strongly suggest the authors do not use 'window' but find another term eg 'trial period' so as to distinguish from real windowed RQA.

For the common use of 'windowed' analysis, see for example:

Webber, C.L., Jr., Marwan, N. editors (2015).Recurrence Quantification Analysis: Theory and Best Practices. Springer Series: Understanding Complex Systems. Springer International Publishing, Cham, Switzerland. http://dx.doi.org/10.1007/978-3-319-07155-8).

The authors are strongly advised to discuss with a statistician how to interpret interactions in regression models.

I will point to the major issues below. I feel I have devoted more than a reasonable amount of time helping improve this manuscript, so will not try to be exhaustive.

LINE 159

H2: The prediction is at odds with the fact that data from in-phase was excluded.

My understanding is that you do not need to compare successful vs unsuccesfull trial to test the hypotheses. So, I don't follow the justification to exclude all in-phase data.

LINE 259:

"350°"

I think you mean "359°"

LINES 269, 272, and others

You say 'categorical CRQA' but in this case the correct comparison would be 'continuous CRQA.'

LINE 295:

"we were able to assess differences in recurrence, predictability, and attractor strength around in-phase, antiphase, and between those regions"

You excluded the data around in-phase (in my view, unnecessarily), so you cannot assess anything about it.

LINES 316, Model specifications.

It is still unclear how the models were fitted.

Please distinguish the variables (success, language experience, tempo range, and 'window') from the possible values (successful/unsuccesful, monolingual/multilingual, low/mid/high, 'window 1' /'window 2'/'window 3'), and indicate how the variables were coded before fitting the model. For example, something like this: trial type was coded as a binary variable with successful = 0 and unsuccessful = 1. Tempo range was coded as a categorical variable using dummy coding...

I looked at the R script and I do not understand what you mean by 'compliant' vs 'noncompliant.' Is this equivalent to successful/unsuccessful? Unclear.

I also believe there may be a serious mistake in how the tempo range data were coded. Specifically, I believe mid-range tempo should probably be coded as lowerTempi == 0 and upperTempi == 0 instead of lowerTempi == -.5 and upperTempi == -.5. Please check.

LINES 339, 348

"standard estimates"

I think you mean 'standardised' not 'standard'

Table 2

The table reports estimates for 'Successful trials' but if 'Successful trials' was the baseline what is this estimate?

The table reports estimates for 'Language experience' but if monolingual language experience was the baseline, this should be multilingual language experience, no?

The table reports estimates for an interaction between "Successful trials" and "Language experience" but if successful and monolingual were the baseline, how do I interpret this interaction effects?

Similar problems for the other interactions (e.g. Successful trials x Lower tempo)

Table 4 - similar issues to Table 2

LINE 352

"This was partially supported by the general MdRQA results for %REC"

No. The results do not partially support H1.

LINE 353

"%REC was significantly greater during middle tempi than lower tempi"

No statistical evidence provided (eg posterior predictions) for this claim. Given the interactions, you cannot simply rely on the 'fixed' (or 'main') effect of "Lower tempo range" to support this claim. Fig 3 also does not provide evidence: the red/blue lines are incorrectly used, given that tempo was binned and coded as categorical.

LINE 355

"Similarly, MAXL was significantly greater during middle tempi than lower tempi and during upper tempi than middle tempi."

No evidence provided. Given the interactions, we cannot infer this from estimates reported in Table 2 alone. And Fig 3C does not help either.

LINE 372

"Trend lines indicate predicted results"

This is not the case, given that tempo was binned into three categories and these lines show a continuous linear increase.

LINE 381

"...include the main effect of window on %REC..."

This is incorrect as it refers only to window 2, and does not consider the interactions.

LINE 384

"The main effect of window on %REC revealed that window 2 had significantly lower %REC than window 1, ..."

As above, given the interactions, this is not granted.

LINE 387

"The interaction between trial type and window on %DET..."

This is only about window 3 not 2. This omission suggests the evidence is stronger than it actually is. And this is in addition to being wrong way of interpreting interactions.

LINE 387

"both Successful and Unsuccessful Trials yielded the highest %DET during window 2 and smallest during window 3"

This is suggested by the summary stats in Table 3. But the link with the results from the regression models is unclear, hence we do not know if these differences are 'significant.'

LINE 391ff

"The interaction between trial type and window on MAXL revealed 391 findings similar to those for %DET. MAXL also peaked during window 2 for Unsuccessful Trials, which supports H2. However, MAXL peaked during window 1 for Successful Trials, opposing H2. Both trial types produced the smallest MAXL values during window 3."

Again, here the results from the regression ('interaction') is offered to support the claim that the differences reported in the summary statistics are significant. But this is not granted.

This problematic interpretation of 'interactions' is repeated multiple times, e.g. L 413, L433, (not exhaustive).

LINE 436

"As measured by %REC, tapping data became less noisy as tempo increased"

This is not supported. This interpretation may be suggested by eyeballing Fig 3, but no statistical evidence is provided, and even eyeballing Fig 3, the difference between monolingual and multilingual seems minimal. Also, to say 'as tempo increased' is not consistent with the fact that tempo was binned and treated as a categorical variable.

LINE 436

"with multilinguals demonstrating a sharper and greater increase than monolinguals."

Not supported

LINE 437

"These findings failed to support H3..."

Indeed the findings contradicted H3.

7. PLOS authors have the option to publish the peer review history of their article (what does this mean?). If published, this will include your full peer review and any attached files.

Reviewer #1: No

Reviewer #2: **Yes: **Murillo Pagnotta

---

## [Author Response · Author response to Decision Letter 1]

18 Oct 2022

Our response to reviewers is attached in the documents section. It's too long to paste here.

---

## [Decision Letter · Decision Letter 2]

3 Nov 2022

PONE-D-21-26808R2Multidimensional recurrence quantification analysis of human-metronome phasingPLOS ONE

Dear Dr. Hall,

also reviewer 2 thinks that the manuscript has improved greatly, and hand only a few final remark where some parts of the manuscript need to be cleaned-up. If you can sumit a final revision that fixes these remaining issues, your article will be accepted for publication.

We look forward to receiving your revised manuscript.

Kind regards,

Sebastian Wallot, Ph.D

Academic Editor

PLOS ONE

Journal Requirements:

Reviewers' comments:

Reviewer's Responses to Questions

**Comments to the Author**

1. If the authors have adequately addressed your comments raised in a previous round of review and you feel that this manuscript is now acceptable for publication, you may indicate that here to bypass the “Comments to the Author” section, enter your conflict of interest statement in the “Confidential to Editor” section, and submit your "Accept" recommendation.

Reviewer #2: (No Response)

2. Is the manuscript technically sound, and do the data support the conclusions?

Reviewer #2: Yes

3. Has the statistical analysis been performed appropriately and rigorously? 

Reviewer #2: Yes

4. Have the authors made all data underlying the findings in their manuscript fully available?

Reviewer #2: Yes

5. Is the manuscript presented in an intelligible fashion and written in standard English?

Reviewer #2: Yes

6. Review Comments to the Author

Reviewer #2: Thank you for the opportunity to revisit this manuscript. The authors have put a lot of effort, and the analyses and their reporting are much improved and much clearer. I am happy to recommend that this manuscript be accepted for publication, and I think the few minor issues below can addressed with the editor directly.

1

the sign for relative phase (psy) is not showing due to formatting to times new roman.

2

the authors added citations to Barr et al 2013 but this work is not included in the reference list.

3

there must have been a mistake along the way when producing the current version. In the response to the reviewer, the authors say they updated line 295 to include this new text:

“In other words, we focused on the systems trajectory after it moved from synchrony, passed through antiphase, and approached synchrony again, rather than focusing on in-phase dynamics.”

But this passage does not appear in the amended manuscript.

4

About figures 3, 4, and 5. In the captions for all three figures, the authors say:

“trend lines indicate predicted results and confidence intervals of the linear mixed effects model, while individual points represent the raw data. Tempo was binned into lower, middle, and upper ranges for the analysis, while the raw data is presented at each individual tempo”

This would be perfect, but the trend lines are still incorrect in all cases. Given that the data was binned, the fitted models will produce predictions for three values of tempo: lower-range, middle-range, and upper-range. However the trend lines in the figures show predictions for 7 values corresponding to the 7 tempos in the raw data (80, 90, 100, 110, 120, 130, 140). It is not possible that the fitted models would have produced this linear increase (unless I am missing something). The authors should amend this figure to correctly represent the fitted models they report.

7. PLOS authors have the option to publish the peer review history of their article (what does this mean?). If published, this will include your full peer review and any attached files.

Reviewer #2: **Yes: **Murillo Pagnotta

---

## [Author Response · Author response to Decision Letter 2]

18 Dec 2022

Our responses are included in the response to reviewers.

---

## [Editor Report · Decision Letter 3]

20 Dec 2022

Multidimensional recurrence quantification analysis of human-metronome phasing

PONE-D-21-26808R3

Dear Dr. Hall,

We’re pleased to inform you that your manuscript has been judged scientifically suitable for publication and will be formally accepted for publication once it meets all outstanding technical requirements.

Kind regards,

Sebastian Wallot, Ph.D

Academic Editor

PLOS ONE
---

## [Editor Report · Acceptance letter]

6 Jan 2023

PONE-D-21-26808R3 

Multidimensional Recurrence Quantification Analysis of Human-Metronome Phasing 

Dear Dr. Hall:

I'm pleased to inform you that your manuscript has been deemed suitable for publication in PLOS ONE. Congratulations! Your manuscript is now with our production department. 

Kind regards, 

on behalf of

Prof. Sebastian Wallot 

Academic Editor

PLOS ONE